# OpenReview forum: "Deep Research Brings Deeper Harm"
_NeurIPS.cc/2025/Workshop/Reliable_ML — NeurIPS 2025 - Reliable ML Workshop_

### Official Review · Reviewer_hSEL · 2025-09-11
**Exposing Alignment Vulnerabilities in Deep Research Agents**

**Rating:** 9
**Confidence:** 3

**Review:**

This paper highlights that the alignment mechanisms of LLMs can be ineffective when models are deployed as Deep Research agents. The authors propose two new jailbreak methods, Plan Injection and Intent Hijack, as well as an evaluation metric, DeepREJECT, to assess the safety of DR agents. They conduct experiments and provide a detailed discussion of why DR agents can bypass alignment safeguards. Overall, this is an interesting and timely work that demonstrates concerning behaviors of DR agents and raises awareness of the associated risks. I believe its quality is suitable for this workshop.

---

### Official Review · Reviewer_Nu3y · 2025-09-19
**Good empirical paper for an workshop**

**Rating:** 7
**Confidence:** 2

**Review:**

Summary
The paper studies adversarial risks of Deep Research agents, showing that standard LLM alignment often fails once wrapped in multi-step search/report pipelines. It introduces two DR-specific jailbreaks: Plan Injection (manipulating plans to add malicious sub-goals) and Intent Hijack (academic reframing of harmful queries). It also proposes DeepREJECT, a new metric combining intent fulfillment and harmful knowledge utility. Experiments across six LLMs and two benchmarks demonstrate DR agents produce more coherent, dangerous outputs than base models.

Strengths
The two jailbreak strategies they tailored are indeed effective across agents.
DeepREJECT captures risks missed by refusal-only metrics.
Solid experiments on multiple models and datasets.

Weaknesses
Intuition behind DeepREJECT not well explained; the choice of  𝛼=0.65, 𝛽=0.35 feels strange.
Attack ideas are not entirely new—similar to prompt injection or H-CoT reframing—though new in DR.

Suggestions
Provide rationale or sensitivity analysis for DeepREJECT’s weighting.
More clearly situate Plan Injection/Intent Hijack relative to prior prompt-injection and reframing work.

Overall
Despite overlaps and missing intuition on DeepREJECT, the evidence is strong and the framing novel for DR agents. I recommend acceptance.